# SEMANTIC-AWARE AND SELF-TRANSFORMATIVE FUNCTION NAME RECOVERY FOR BINARIES

## ABSTRACT

Reverse engineers aim to analyze stripped binaries in order to identify and mitigate software vulnerabilities. Unlike source code, real-world binaries contain limited semantic information, as companies often remove symbols to reduce file size and protect intellectual property. This lack of information makes program comprehension challenging. Since binaries consist of numerous functions, recovering meaningful function names is a crucial step toward understanding program behavior. Recent work has applied machine learning to this task, and Large Language Models (LLMs) have shown promise due to their ability to generate contextually relevant identifiers beyond the training set. However, progress is hindered by limited training data, underscoring the need for optimized fine-tuning strategies. To address this, we propose SYMSEM, a self-transformative, semantic-aware fine-tuning framework for function name recovery. We evaluate SYMSEM across three architectures, x86-64, ARM, and MIPS, and demonstrate that it significantly outperforms prior approaches, achieving up to 68% higher F1 score on MIPS compared to the state-of-the-art.

## 1 INTRODUCTION

Software has become ubiquitous worldwide, driven by rapid advances in computer science. However, software is inherently prone to vulnerabilities, which can lead to severe consequences such as privilege escalation or remote code execution in modem system (pri). To mitigate these risks, reverse engineering has become a key technique for analyzing released binaries and identifying potential weaknesses. Beyond its defensive role, reverse engineering can obtain a benefit by the security industry researchers can earn rewards of up to two million dollars for discovering critical bugs in Apple products (bug). Among reverse engineering tasks, function name recovery is particularly important because it provides high-level insights into program functionality, thereby improving both vulnerability discovery and program comprehension.

Despite its usefulness, function name recovery remains a challenging problem due to several factors. First, stripped binaries typically omit symbol information, both to reduce file size and to hinder reverse engineering efforts. Second, compiler optimizations, such as inlining, reordering, or dead-code elimination, distort the original structure, making it harder to infer meaningful function names. Third, binaries can be represented in different forms, such as assembly code or decompiled code, each with its own trade offs. Assembly code has a limited vocabulary but conveys little semantic meaning, while decompiled code is more human-readable but introduces an unbounded vocabulary, complicating recovery.

To overcome these challenges, recent work has explored the use of Large Language Models (LLMs). For example, SYMGEN (Jiang et al., 2025) leverages decompiled code along with function summaries generated by code LLMs for fine-tuning. This summary-based approach improves performance, particularly for previously unseen functions, by exploiting the generative capabilities of LLMs.

However, SYMGEN has several limitations. It requires function summaries, which in turn demand additional preprocessing to extract and generate. The quality of these summaries heavily depends on the LLM used, making the approach sensitive to external factors. Moreover, generating summaries for new datasets introduces significant computational overhead. Finally, relying primarily on sum-

maries overlooks rich semantic information already embedded in decompiled code, such as calling conventions and variable information.

To address these challenges, we propose SYMSEM, a framework that leverages a semantic-aware, self-transformative fine-tuning process. SYMSEM operates in two stages: 1) Fine-tuning the model to recover variable names, thereby enriching the semantic context of functions, and 2) Fine-tuning the model to recover function names, while simultaneously generating multiple representations from a single decompiled function to enable the LLM to capture deeper semantic meaning. The results from the first stage are incorporated during testing of the second-stage model, providing additional semantics that further improve recovery accuracy.

We evaluate SYMSEM across three architectures, x86-64, ARM, and MIPS, under four compiler optimization levels (O0-3). Experimental results demonstrate that SYMSEM consistently outperforms SYMGEN on all architectures. Most notably, SYMSEM achieves around an 68% higher F1 score on MIPS compared to the state-of-the-art.

**Contributions.** This work makes the following contributions:

- We propose SYMSEM, a novel and lightweight framework for function name recovery from stripped binaries.
- We design a semantic-aware two-stage pipeline that integrates variable recovery with a self-transformative dataset expansion strategy, enabling robust fine-tuning under limited data.
- We perform comprehensive cross-architecture evaluation across x86-64, ARM, and MIPS, demonstrating that SYMSEM consistently outperforms prior work, with particularly substantial improvements on the MIPS architecture.

## 2 BACKGROUND

**Compilation.** When programmers write a C program, it must be compiled into an executable binary before it can run on a target system. During this process, developers must decide the intended deployment environment. For instance, software designed for low-power embedded systems may be compiled for an ARM architecture rather than x86-64. Each architecture defines a distinct instruction set, and this choice is fixed at compilation time.

In addition to selecting the architecture, programmers must specify the optimization level, which determines how aggressively the compiler transforms source code into machine code. Optimization levels balance execution speed against binary size, and most compilers offer four standard settings (O0-3). At O0, no optimizations are applied, while O3 enables aggressive transformations to maximize performance and minimize size.

Finally, programmers may decide whether to include *debug symbols*, which preserve high-level information such as function and variable names. Binaries compiled with debug symbols are called *unstripped binaries*. Although they facilitate debugging and program analysis, unstripped binaries are significantly larger and may reveal sensitive intellectual property. For this reason, most production software is distributed in a stripped form, where debug symbols are removed. *Stripped binaries* are smaller and more secure for release, but they lack critical semantic information.

**Decompilation.** Reverse engineers typically analyze stripped binaries to understand their functionality and detect vulnerabilities. The most common technique is decompilation, which employs tools such as Ghidra (Agency) or IDA Pro (hex rays) to reconstruct machine code into a higher-level, human-readable representation known as decompiled code. While decompilation produces pseudocode that resembles C source code, stripped binaries make the process considerably more difficult. In the absence of debug symbols, variable and function names are replaced by automatically generated placeholders, and other semantic information lost during compilation cannot be fully recovered. As a result, decompiled code is often incomplete or ambiguous, making reverse engineering both time-consuming and error-prone.

## 3 RELATED WORKS

**Function Name Recovery.** Function name recovery is the task of inferring the original function name given only the decompiled code. Several approaches have been proposed in recent years. NERO (David et al., 2020) employs control-flow graphs in combination with transformer architectures (Vaswani et al., 2017) to recover function names. SYMLM (Jin et al., 2022) also adopts transformer models but incorporates function context, capturing the execution behavior of a given function. ASMDEPCITOR (Kim et al., 2023) encodes binary function semantics directly from assembly code using a transformer-based model. These three works primarily rely on assembly code as the representation for recovery.

Moving beyond raw assembly, XFL (Patrick-Evans et al., 2023) integrates multiple semantic signals, including call graphs and global binary context. Instead of relying solely on assembly code, XFL leverages an intermediate representation (IR) to construct function embeddings. More recently, SYMGEN (Jiang et al., 2025) introduced the use of LLMs for function name recovery. Unlike prior assembly-based and IR approaches, SYMGEN operates on decompiled code and enhances fine-tuning with automatically generated function summaries, which improve generalization, particularly for previously unseen functions.

**Variable Information Recovery.** In parallel, a line of research has focused on recovering variable-related information. OSPREY (Zhang et al., 2021) formulates probabilistic constraints to infer variable types. DIRTY (Chen et al., 2022) employs a transformer-based neural network to recover variable names and types. RESYM (Xie et al., 2024) leverages LLMs to recover both variable names and data structures. VARBERT (Pal et al., 2024) applies transfer learning for variable recovery, while GENNM (Xu et al., 2025) introduces a generative modeling approach. Although variable information recovery has value on its own, it is not our primary objective. In SYMSEM, we employ variable recovery only as a pre-processing step to enrich stripped decompiled code, thereby improving the accuracy of function name recovery.

## 4 MOTIVATION

Despite recent progress, existing approaches to function name recovery still face fundamental limitations that hinder their effectiveness. We focus on the state-of-the-art method, SYMGEN (Jiang et al., 2025), to illustrate these challenges, and then present two insights that motivate our proposed solution.

**Background on** SYMGEN**.** SYMGEN fine-tunes a model by collecting decompiled code from binaries with debug symbols, ensuring that the input code contains the original symbol information. It then augments this dataset with function summaries generated by code LLMs. Finally, the model is fine-tuned jointly on decompiled code and summaries. While this approach improves generalization, particularly for unseen functions, it introduces several drawbacks.

**C1: Ineffectiveness of Summary Generation.** SYMGEN relies on function summaries produced by LLMs, but these are prone to hallucination and often contain misleading or incorrect descriptions. There is no reliable mechanism to verify their accuracy, and source code comments, which might otherwise serve as ground truth, are rarely available or consistent. Moreover, summary generation introduces significant computational overhead, since new summaries must be created for each dataset. This makes the process resource-intensive, error-prone, and difficult to scale.

**C2: Limited Availability of Data.** Function name recovery requires binaries with preserved symbols or access to paired source code, yet such resources are scarce. Although vast repositories exist on GitHub, there is no guarantee that function names remain consistent with compiled binaries. To ensure reliability, prior works limit training to well-established projects across diverse domains, but this drastically reduces dataset diversity. While cross-compilation can extend coverage across architectures, the scarcity of symbol-preserving binaries persists, leaving LLMs with insufficient data for effective fine-tuning.

**C3: Loss of Semantics in Decompiled Code from Stripped Binaries.** Decompiler-based approaches attempt to leverage contextual information such as variable names and callee functions.

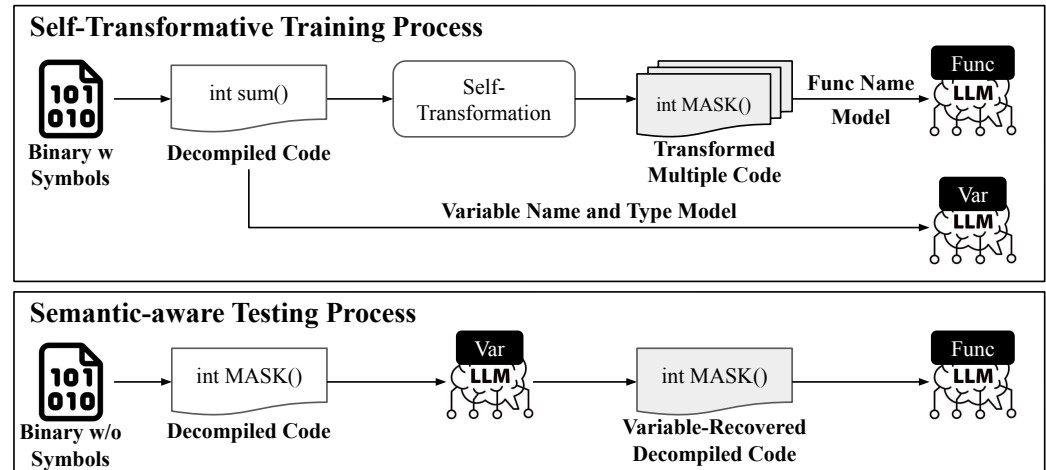

**Figure 1:** SYMSEM **Overview.**

However, these semantics depend heavily on the presence of debug symbols, which are absent in real-world binaries. As a result, test data often lack critical semantic information, creating a semantic gap between training and inference. This mismatch limits the ability of LLMs to generalize and accurately recover function names.

**S1: Self-Transformative Data.** Since function summaries are incomplete and unreliable, we discard them. However, removing summaries reduces the dataset size, as each function yields only one sample instead of both a function and its summary. To overcome this limitation, we propose a self-transformative approach that generates multiple alternative representations from a single decompiled code. This expands the dataset, exposes the model to diverse semantic perspectives, and enhances the robustness of fine-tuning without the need for costly summaries.

**S2: Variable Information Utilization.** Real-world test binaries lack debug symbols, and thus their decompiled functions are stripped of variable semantics. To mitigate this, we fine-tune a dedicated LLM for variable information recovery, including variable names and types. During inference, this model enriches stripped decompiled functions with recovered variable semantics. The enriched code is then fed into the function name recovery model, narrowing the semantic gap between training and testing and significantly improving recovery accuracy.

## 5 SYMSEM

### 5.1 OVERVIEW

As illustrated in Figure 1, SYMSEM is designed as a two-stage framework centered on LLMs, targeting both variable information recovery and function name recovery. During fine-tuning, we use binaries compiled with debug symbols to construct datasets for both tasks. In the first stage, an LLM is fine-tuned to recover variable information from decompiled functions, while in parallel another LLM is fine-tuned to recover function names.

At inference time, the pipeline operates exclusively on stripped binaries. The variable information model (VAR-LLM) first enriches decompiled code by predicting variable identifiers. The enriched code is then passed to the function name model (FUNC-LLM), which generates the final function name prediction. Through this semantic enrichment and hierarchical recovery process, SYMSEM narrows the semantic gap between training and testing conditions.

## 5.2 DATASET VIA SELF-TRANSFORMATIVE APPROACH

**Data Preparation.** We collect large-scale source code projects and compile them across multiple architectures and optimization levels, following datasets used in prior work (Jin et al., 2022; Jiang et al., 2025). Each compilation produces both unstripped binaries and stripped binaries. Unstripped binaries provide ground-truth labels such as function names, variable names, and variable types. Stripped binaries, which resemble real-world deployment scenarios, serve as inputs but lack annotations.

**Data Generation.** We generate decompiled code using Ghidra (Agency). Our data generation pipeline extends SYMGEN 's implementation. Since variable information is required for VAR-LLM, we extract ground-truth variable details directly from unstripped binaries. Ghidra provides the stack locations of local variables and register information for function parameters, which allow us to differentiate variables accurately. Using this information, we annotate the stripped binary functions with variable names and types, producing parallel datasets of stripped and unstripped decompiled code along with their corresponding labels. As a last stage, we replace the function name with the mask token (i.e., MASK) to make LLMs easy to perform function name recovery task. Otherwise, the function name will be replaced by a placeholder defined by Ghidra, which makes inefficient.

**Deduplication.** We adopt the deduplication strategy proposed by the previous work (Jiang et al., 2025) to ensure data quality. Decompiled code often resembles C source code but may vary across projects due to autogenerated identifiers or differing function addresses. To eliminate redundancy, we normalize decompiler-specific tokens and then construct abstract syntax trees (ASTs) for structural comparison. Functions that are structurally identical at the AST level are removed. Additionally, functions with overly frequent or repeated names are pruned to reduce label imbalance and prevent the model from being biased toward trivial identifiers.

**Self-Transformative Approach.** At this stage, we leverage both stripped and unstripped decompiled functions, which originate from the same source. However, the dataset size remains limited compared to SYMGEN. To address this, SYMSEM introduces a self-transformative mechanism that systematically expands the data.

In practice, the format of test data for SYMSEM consists of decompiled code from stripped binaries enriched with recovered variable information. To simulate this during training, we generate multiple modified versions of each function by gradually reintroducing variable semantics. For example, given a decompiled function with ten variables, the stripped version lacks any semantic details. SYMSEM may first replace two variables (20%) with their ground-truth names and types, and then progressively replace the remainder. This process generates multiple semantically distinct variants, capturing scenarios where variable recovery is partial or imperfect. The replacement ratio can be tuned; for instance, a 25% ratio produces four additional variants, yielding six distinct versions of the same function.

Replacing variables in decompiled code is a non-trivial task due to decompiler-specific pseudo-code representations and the inherent complexity of code formatting. One possible approach is to leverage decompiler APIs to substitute variables with their corresponding names and types, and then re-generate the decompiled code every time. However, this method is computationally expensive, as it requires the decompiler to re-analyze the binary each time a modification is made. To avoid this overhead, we instead operate directly on the AST generated with ANTLR 4 (Parr, 2013). Within the AST, nodes corresponding to variable definitions are identified and selectively replaced with ground-truth names and types, with the changes consistently propagated throughout the function body. This strategy enables the efficient generation of multiple semantically valid variants while preserving structural correctness and eliminating the need for repeated decompilation.

## 5.3 TRAINING LLM

**Parameter-Efficient Fine-tuning.** Given the large parameter count of LLMs and limited resources, we employ parameter-efficient fine-tuning methods. Specifically, we use low-rank adaptation (LoRA) (Hu et al., 2022; Dettmers et al., 2023), which freezes original model weights while learning low-dimensional trainable matrices via backpropagation. Moreover, since it is not neces-

sary to train every single layer from the model, it limits the layers for the fine-tuning process. As a result, for the CodeLlama-34B model, this reduces the number of trainable parameters by approximately 99.97%, significantly lowering computational cost and enabling efficient fine-tuning.

**Training Objective.** Since both VAR-LLM and FUNC-LLM follows the fine-tuning process with the identical objective, we illustrate this before describing VAR-LLM and FUNC-LLM. Formally, both VAR-LLM and FUNC-LLM are trained as autoregressive generation models. Given an input sequence $\mathbf{x} = \{x_1, x_2, x_3, ..., x_n\}$ and output sequence $\mathbf{y} = \{y_1, y_2, y_3, ..., y_m\}$, SYMSEM maximizes the following log-likelihood:

$$\max_\theta \log P_\theta(y) = \sum_{i=1}^{m} \log P_\theta(y_i \mid x; y_{1:i-1}) \tag{1}$$

$$\log P_\theta(y_i \mid x; y_{1:i-1}) = \frac{\exp(h_\theta(x; y_{1:i-1})^\top e(y_i))}{\sum_{y'} \exp(h_\theta(x; y_{1:i-1})^\top e(y'))}, \tag{2}$$

where $h_\theta(\cdot)$ denotes the hidden state representations from intermediate layers and $e(\cdot)$ the input embedding.

**Variable Information Recovery Model (VAR-LLM).** We fine-tune the model using the Alpaca-LoRA (tloen) platform. Input data consists of stripped decompiled code only with the corresponding variable ground truth information. Since functions often contain multiple variables, we do not mask variables individually; instead, we provide explicit variable name and type annotations to guide the model. The ground truth for variable information contains two components following the format, such as "param_1: int, argc", capturing both name and type.

**Function Name Recovery Model (FUNC-LLM).** FUNC-LLM is fine-tuned in a similar manner, but trained on self-transformative datasets to capture diverse semantic contexts. This allows the model to generalize better across functions with incomplete or partially recovered semantics.

### 5.4 RECOVERING FUNCTION NAME

At inference, we apply the two fine-tuned models sequentially. First, VAR-LLM takes fully stripped decompiled code as input and predicts variable information. These predictions, even if imperfect, are integrated into the decompiled code to enrich its semantics. The identical replacement algorithm used in self-transformative approach ensures consistent updates throughout the function body. The enriched function is then passed to FUNC-LLM, which generates the final function name.

**Correctness Determination.** Since function naming conventions vary (e.g., `clear` vs. `clr`), exact string matching is insufficient. We adopt CodeWordNet (Jin et al., 2022), which decomposes function names into subwords and computes distances between them, allowing partial matches to be considered partial correct, which provides scores depending on the number of matches. Finally, to ensure deterministic evaluation, we disable randomization parameters such as temperature, top-p, and top-k during inference.

## 6 EXPERIMENTS

### 6.1 DATASETS

Our dataset covers three architectures, x86-64, ARM 32-bit, and MIPS 32-bit, each compiled under four optimization levels (O0-3) To ensure comparability, we adopt the same dataset setup as SYM-GEN (Jiang et al., 2025) for the fair comparison. In this setup, projects are partitioned into training (10%), validation (10%), and test (80%) sets, with the split determined at the project level rather than by function count. However, due to varying project sizes and the deduplication process, this results in imbalanced distributions of functions across splits. To mitigate this issue, we reorganize the datasets such that functions themselves are distributed according to the same ratio, ensuring a more balanced and reliable evaluation. As a result, even though two works share the identical datasets, the component of train, validation, and test sets is different.

| Arch | Model | O0 | | | O1 | | | O2 | | | O3 | | |
|---|---|---|---|---|---|---|---|---|---|---|---|---|---|
| | | Prec | Recall | F1 | Prec | Recall | F1 | Prec | Recall | F1 | Prec | Recall | F1 |
| x86-64 | ASMDEPICTOR | 0.070 | 0.090 | 0.079 | 0.078 | 0.103 | 0.089 | 0.079 | 0.096 | 0.087 | 0.079 | 0.105 | 0.090 |
| | SYMLM | 0.183 | 0.056 | 0.086 | 0.187 | 0.063 | 0.094 | 0.188 | 0.065 | 0.097 | 0.213 | 0.083 | 0.120 |
| | xFL | 0.080 | 0.095 | 0.087 | 0.071 | 0.085 | 0.077 | 0.066 | 0.071 | 0.069 | 0.088 | 0.095 | 0.091 |
| | SYMGEN | 0.263 | 0.285 | 0.273 | 0.264 | 0.290 | 0.275 | 0.282 | 0.311 | 0.296 | 0.283 | 0.310 | 0.296 |
| | SYMSEM | 0.271 | 0.297 | 0.284 | 0.276 | 0.276 | 0.276 | 0.302 | 0.316 | 0.309 | 0.314 | 0.319 | 0.317 |
| | vs. SYMGEN | ↑3.04% | ↑4.21% | ↑4.02% | ↑4.54% | ↓4.83% | ↑0.36% | ↑7.09% | ↑1.60% | ↑1.04% | ↑10.95% | ↑2.90% | ↑7.09% |
| ARM | ASMDEPICTOR | 0.045 | 0.052 | 0.048 | 0.045 | 0.048 | 0.047 | 0.040 | 0.041 | 0.041 | 0.036 | 0.041 | 0.039 |
| | SYMLM | 0.080 | 0.021 | 0.033 | 0.045 | 0.016 | 0.024 | 0.042 | 0.018 | 0.026 | 0.035 | 0.015 | 0.021 |
| | SYMGEN | 0.238 | 0.237 | 0.238 | 0.264 | 0.244 | 0.254 | 0.251 | 0.238 | 0.244 | 0.254 | 0.234 | 0.244 |
| | SYMSEM | 0.251 | 0.235 | 0.243 | 0.278 | 0.259 | 0.269 | 0.267 | 0.256 | 0.262 | 0.258 | 0.256 | 0.257 |
| | vs. SYMGEN | ↑5.46% | ↓0.84% | ↑2.10% | ↑5.30% | ↑6.14% | ↑5.90% | ↑6.37% | ↑7.56% | ↑7.37% | ↑1.57% | ↑9.40% | ↑5.32% |
| MIPS | ASMDEPICTOR | 0.056 | 0.049 | 0.053 | 0.034 | 0.041 | 0.037 | 0.050 | 0.049 | 0.050 | 0.047 | 0.042 | 0.044 |
| | SYMLM | 0.133 | 0.033 | 0.053 | 0.103 | 0.025 | 0.040 | 0.138 | 0.036 | 0.057 | 0.112 | 0.026 | 0.043 |
| | SYMGEN | 0.187 | 0.183 | 0.185 | 0.205 | 0.207 | 0.206 | 0.201 | 0.201 | 0.201 | 0.191 | 0.189 | 0.190 |
| | SYMSEM | 0.209 | 0.208 | 0.209 | 0.276 | 0.282 | 0.279 | 0.275 | 0.272 | 0.274 | 0.331 | 0.311 | 0.321 |
| | vs. SYMGEN | ↑11.76% | ↑13.66% | ↑12.97% | ↑34.64% | ↑36.23% | ↑35.43% | ↑36.81% | ↑35.32% | ↑36.31% | ↑73.29% | ↑64.55% | ↑68.94% |

**Table 2: Performance Evaluation of Every Work on Function Name Recovery.**

In addition, we apply self-transformation with a granularity of 25%. This means that for each stripped decompiled function, four additional transformed variants are generated (25%, 50%, 75%, and 100% variable replaced), resulting in five distinct versions per function.

## 6.2 EVALUATION

**LLMs.** We use CodeLlama 34B (Roziere et al., 2023) as the base model for both VAR-LLM and FUNC-LLM. This choice follows the prior work (Jiang et al., 2025), which confirmed that the dataset used in our experiments does not overlap with CodeLlama's training data, ensuring a fair evaluation and prevention of data leakage.

**Prompt settings.** For variable information recovery, we adopt the prompt from RESYM (Xie et al., 2024), *"What are the original name and data type of variables param_1, param_2?"*. For function name recovery, we adopt the prompt used by SYMGEN, *"Suppose you are an expert in software reverse engineering. Here is a piece of decompiled code, you should infer code semantics and tell me the original function name from the contents of the function to replace [MASK]. Now the decompiled codes are as follows:"*.

**Baselines.** We compare SYMSEM with four existing approaches, SYMGEN (Jiang et al., 2025), ASMDEPICTOR (Kim et al., 2023), SYMLM Jin et al. (2022), and xFL Patrick-Evans et al. (2023). Since xFL only supports the x86-64 architecture, we restrict its comparison to that platform, while the others are evaluated across all three architectures. As our focus is function name recovery, we do not report variable recovery performance comparisons with works such as RESYM (Xie et al., 2024) and VARBERT (Pal et al., 2024).

**Evaluation metrics.** Following prior work (Jin et al., 2022; Jiang et al., 2025), we evaluate using precision, recall, and F1 score, which together provide a comprehensive measure of accuracy and robustness.

## 6.3 MAIN RESULTS

**VAR-LLM Results.** Before evaluating SYMSEM as a whole, we first assess the performance of VAR-LLM on variable recovery, since its predictions are used to enrich the inputs for FUNC-LLM. Results are calculated using CodeWordNet (Jin et al., 2022), which measures similarity based on subword decomposition. Table 1 summarizes the outcome.

Overall, the accuracy of VAR-LLM is lower than specialized approaches such as RESYM (Xie et al., 2024), which reports 37.5% accuracy under a different dataset. Among the three architectures, VAR-LLM achieves slightly higher F1 scores on x86-64 compared to ARM and MIPS. This result indicates that even though variable recovery performance is not completely accurate, it could provide additional semantic signals for FUNC-LLM due to the correctly inferred variable names.

| Arch | F1 Score | | | |
|---|---|---|---|---|
| | O0 | O1 | O2 | O3 |
| x86-64 | 0.263 | 0.189 | 0.202 | 0.222 |
| ARM | 0.174 | 0.157 | 0.150 | 0.148 |
| MIPS | 0.102 | 0.194 | 0.199 | 0.200 |

**Table 1: VAR-LLM Performance on Variable Name Recovery.**

FUNC-LLM **Results.** Table 2 compares SYMSEM with existing baselines. Overall, SYMSEM and SYMGEN are the only methods that consistently achieve F1 scores above 0.2, whereas ASMDE-PICTOR, SYMLM, and xFL struggle to handle unseen functions. For example, none of these three methods exceeds 0.1 F1 on the x86-64 O0 setting. Interestingly, while xFL reports an F1 of 0.241 on unseen functions in its original paper, under our experimental conditions it achieves only 0.087, which is only one-third.

A key limitation of these baselines, with the exception of SYMGEN, is that they can only generate tokens observed during training. Several rely on subword tokenization that they decompose function names into subwords, train on those units, and then attempt to reconstruct unseen names via previously built subwords. However, when a new function name consists of subwords not encountered during training, these models fail to generalize. Covering all possible subwords would require enormous datasets, making such methods inefficient. In contrast, SYMSEM and SYMGEN leverage the generative capacity of LLMs, enabling them to produce novel function names beyond the training vocabulary.

On x86-64, both SYMSEM and SYMGEN achieve competitive F1 scores, with SYMSEM slightly outperforming SYMGEN across all optimization levels. More importantly, SYMGEN's performance declines notably on ARM and MIPS. For instance, on O0 optimization, SYMGEN achieves F1 scores of 0.238 on ARM and 0.185 on MIPS, compared to 0.273 on x86-64. This suggests that the base model is less familiar with underrepresented architectures such as MIPS, whose popularity has declined in recent years. In contrast, SYMSEM maintains stronger performance across all architectures, with particularly large improvements on MIPS. Table 2 also describes the direct comparison between SYMSEM and SYMGEN, to clearly illustrate the improvement.

We also observe a correlation between VAR-LLM and FUNC-LLM performance. Architectures where VAR-LLM performs better tend to yield higher function name recovery scores. For example, SYMSEM achieves its highest function recovery on MIPS O3, where VAR-LLM also performs relatively well, which is 0.2 F1 score. Conversely, SYMSEM's lowest performance on MIPS O0 aligns with the lowest VAR-LLM score (0.102). While the relationship is not strictly linear, these results highlight that improvements in variable recovery positively influence downstream function name recovery.

## 6.4 ABLATION RESULTS

SYMSEM allows different ratios of variable replacement during self-transformative process. To evaluate the effect of this parameter, we conduct experiments by varying the replacement ratio. In addition, we apply the same strategy of SYMSEM to SYMGEN to test its effectiveness under comparable settings. Finally, we investigate the impact of variable recovery accuracy by substituting predicted variables with ground-truth values.

**Different Ratio of Variable Replacement without Summary.** We first vary the replacement ratio for self-transformation on the MIPS O3 setting, which exhibits the largest performance gap between SYMSEM and SYMGEN and is therefore most suitable for ablation testing. Including SYMSEM's default ratio 25%, we additionally test 50% and 100% replacements. A ratio of 50% means that half and all of the variables are replaced during fine-tuning, while 100% means that all variables are replaced.process. A lower ratio naturally produces more augmented samples, as multiple variants can be generated from the same function.

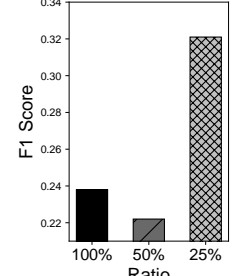

**Figure 2: F1 Scores with Different Ratios of Variable Replacement.**

Figure 2 shows the results. The best performance is achieved with a 25% ratio, which is the method SYMSEM applied, while both 50% and 100% produce slightly lower F1 scores. Interestingly, the 50% ratio performs worse than 100%, suggesting that replacing half the variables does not provide additional useful training signals. This may be because variables were randomly selected for replacement, and thus the added variants were less informative. Nonetheless, the performance differences remain relatively small, confirming the robustness of the self-transformative strategy.

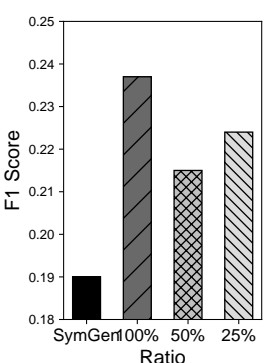

**Figure 3: F1 Scores with Different Ratios of Variable Replacement with Summary.**

**Different Ratio of Variable Replacement with Summary.** Next, we apply the self-transformative and semantic-aware method to SYMGEN to assess its impact when summaries are included. Similar to the previous experiment, we test ratios of 25%, 50%, and 100% on MIPS O3. The results, shown in Figure 3, reveal two important findings. First, all self-transformative variants outperform the original SYMGEN, which does not apply both self-transformative and sematic-aware methods, confirming the effectiveness of the method. Second, consistent with the previous experiment, fine-grained ratios do not always yield better results. The 100% ratio achieves the highest F1 score, followed by 25%, while 50% again performs worse than expected.

More importantly, across all ratios, SYMGEN with summaries performs worse than the corresponding self-transformative variants without summaries, presented in Figure 2. This indicates that summaries may actually hinder performance, reinforcing our motivation to eliminate them in SYMSEM. Thus, self-transformation proves not only effective but also more reliable than summary-based augmentation, with the choice of ratio being the more critical factor.

**Variable Ground Truth Information Utilization.** Finally, we investigate the relationship between VAR-LLM accuracy and FUNC-LLM performance by replacing all variables in the decompiled code with their ground truth names and types. This experiment represents an upper bound where variable recovery achieves perfect accuracy (i.e., F1 score is 1.0). We again use the MIPS O3 setting, where SYMSEM already achieves strong results.

As shown in Figure 4, substituting ground-truth variables increases the F1 score from 0.32 to 0.35, an improvement of roughly 10%. While this demonstrates that more accurate variable recovery directly benefits function name recovery, the improvement is relatively modest. This finding highlights the robustness of SYMSEM's design: despite the limited accuracy of VAR-LLM, SYMSEM already achieves substantial gains, and further improvements in variable recovery only enhance performance incrementally.

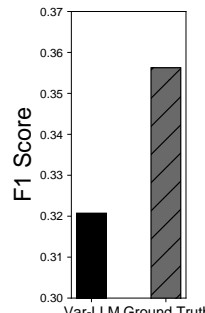

**Figure 4: F1 Scores with VAR-LLM Replacement and Ground Truth.**

## 7 FUTURE WORKS & CONCLUSION

**Future Works.** As future work, we plan to develop more advanced techniques for variable information recovery, aiming to enhance both accuracy and robustness. In addition, we will conduct more extensive experiments to systematically determine the optimal ratio of variable replacement in the self-transformative process. This will allow us to refine the balance between data augmentation and semantic fidelity.

**Conclusion.** We introduce SYMSEM, a novel framework for function name recovery from stripped binaries that integrates semantic-aware fine-tuning with a self-transformative data augmentation strategy. SYMSEM enriches decompiled code with variable information, such as names and types, thereby narrowing the semantic gap between training and testing. To address the limited availability of training data, SYMSEM generates multiple semantically consistent variants of each function through controlled variable replacement. We evaluate SYMSEM across three architectures under four optimization levels. Experimental results demonstrate that SYMSEM consistently outperforms existing approaches, achieving up to 68% higher F1 score on MIPS compared to the state-of-the-art method, SYMGEN. These findings highlight the effectiveness of our semantic-aware and self-transformative design in improving the generalization of LLMs for binary analysis.

**Reproducibility.** All experiments can be reproduced using the provided scripts. Minor variations in reported results may occur due to randomness introduced during fine-tuning and the self-transformative augmentation process.

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

## A LLM USAGE

We deploy LLM for correcting and polishing the paragraphs.