# OpenReview forum: "Semantic-Aware and Self-Transformative Function Name Recovery for Binaries"
_ICLR.cc/2026/Conference — ICLR 2026 Conference Withdrawn Submission_

### Official Review · Reviewer_2hF1 · 2025-10-25

**Soundness:** 2
**Presentation:** 2
**Contribution:** 2
**Rating:** 2
**Confidence:** 4

**Summary:**

This paper addresses the challenge of recovering meaningful function names from stripped binaries—an essential task for reverse engineering and vulnerability analysis. Real-world binaries lack symbolic and semantic information, making program comprehension difficult. While recent machine learning and large language model (LLM)-based methods have improved function name recovery, their effectiveness is limited by insufficient training data and suboptimal fine-tuning strategies. To overcome these limitations, the authors introduce SymSem, a self-transformative, semantic-aware fine-tuning framework designed to enhance LLM performance on this task. SymSem leverages semantic understanding to recover function names more effectively across different instruction set architectures. Experimental results on x86-64, ARM, and MIPS binaries show that the proposed method substantially outperforms prior work, achieving up to a 68% improvement in F1 score on MIPS. This demonstrates SymSem’s capability to improve program understanding and advance automated reverse engineering.

**Strengths:**

The paper explores MIPS binaries, which less explored in the field.

**Weaknesses:**

The paper’s motivation appears to rely primarily on SYMGEN, with limited discussion or critical analysis of other prior studies in the field. It does not clearly articulate the limitations of existing approaches beyond SYMGEN or explain the broader motivation for this work beyond serving as an incremental improvement. Additionally, the concept of refining variable names prior to function name generation is already well established; therefore, the paper should clarify the novel contribution and how it differentiates from existing methods using similar strategies.

**Questions:**

the paper need significant improvement.

---

### Official Review · Reviewer_ntW2 · 2025-10-26

**Soundness:** 1
**Presentation:** 2
**Contribution:** 2
**Rating:** 2
**Confidence:** 3

**Summary:**

The paper proposes a new framework called SYMSEM for recovering function names from stripped binaries. The paper point out that existing SOTA methods (e.g., SYMGEN) rely on LLM-generated function summaries for fine-tuning, but these summaries can contain hallucinations, be inaccurate, and are costly to generate. To address these issues, SYMSEM introduces a two-stage, semantic-aware, and self-transformative framework: (i) Semantic-Aware: The framework first trains a VAR-LLM to recover variable names and types from the binary code. At test time, this model first "enriches" the stripped decompiled code with the predicted variable information. (ii) Self-Transformative: To tackle the problem of limited training data, the paper propose a data augmentation strategy. Instead of using function summaries, this strategy generates multiple training variants for a single function by progressively reintroducing different ratios of ground-truth variable information into the stripped code. The "enriched" code is then fed into a second model, FUNC-LLM, to predict the final function name. The paper conduct experiments on three architectures (x86-64, ARM, and MIPS) and claim their method significantly outperforms prior work, particularly on the MIPS architecture, "achieving up to 68% higher F1 score on MIPS compared to the state-of-the-art".

**Strengths:**

This paper addresses the important and challenging problem of recovering function names from stripped binaries. The main strength of the paper lies in its motivation: the authors insightfully identify the risks associated with existing SOTA methods (like SYMGEN), which rely on LLM-generated summaries for training, pointing to issues of hallucination and high cost. The paper attempts to replace this potentially unreliable method with a more controllable and lower-cost strategy, namely, using variable information for "self-transformative" data augmentation. This approach is reasonable and valuable. Furthermore, the paper includes ablation studies to validate its internal framework design choices (e.g., different variable replacement ratios).

**Weaknesses:**

1.Regarding C3 in the motivation (section 4), when the authors describe the shortcomings of the SYMGEN paper, they mention, "Decompiler-based approaches attempt to leverage contextual information such as variable names and callee functions." and "test data often lack critical semantic information, creating a semantic gap between training and inference.". I believe the authors have misunderstood SYMGEN here. In SYMGEN, the input is always stripped binaries during both training and inference; therefore, no such "semantic gap between training and inference" exists.

2.The authors first state that they adopt the same dataset setup as SYMGEN, where projects are partitioned into training (10%), validation (10%), and test (80%) sets. This is the correct methodology for evaluating a code model's ability to generalize to unseen projects. However, the authors then state they changed this setup: "To mitigate this issue, we reorganize the datasets such that functions themselves are distributed according to the same ratio, ensuring a more balanced and reliable evaluation.". This change in data partitioning (from project-level to function-level) is highly problematic. It very likely introduces severe data leakage. Functions from the same project are likely to share similar naming conventions, styles, and code structures. By sampling randomly at the function level, it is almost certain that the test set includes functions from projects that are already present in the training set. This data leakage simplifies the test task, as the model no longer needs to generalize to new projects, but merely to unseen functions from the same projects.

3.The proposed SYMSEM is a modular, two-stage framework, with the core idea of using variable information (Stage 1, VAR-LLM) to guide function name recovery (Stage 2, FUNC-LLM). However, in the experiments (Section 6.3), the paper states, "the accuracy of VAR-LLM is lower than specialized approaches such as RESYM (Xie et al., 2024), which reports 37.5% accuracy under a different dataset.". Given that this component performs worse than SOTA algorithms, a SOTA algorithm could have been used to perform the variable information recovery task. The introduction of this underperforming VAR-LLM seems unnecessary.

4.The paper lacks novelty. The main contributions are presented as: (i) the "semantic-aware" approach, i.e., using function variable information to assist in function name recovery, and (ii) the "self-transformative dataset expansion strategy." Regarding (i), the SYMGEN paper already mentioned in its 'FURTHER ANALYSIS' section (and supported with experiments) that variable information can effectively aid in function name recovery. Moreover, the variable recovery method proposed in this paper does not perform well. Therefore, this cannot be considered a novel contribution. Regarding (ii), this is merely a data augmentation technique. This alone is insufficient to be a major contribution.

5.The experimental evaluation is insufficient. The paper's evaluation is limited to its academic test set and completely fails to assess its robustness in real-world binary analysis scenarios, such as its performance on Obfuscated Binaries, Malware, and IoT Firmware.

**Questions:**

Q1: Can you provide evidence that your function-level dataset split does not suffer from the data leakage issues outlined in the weaknesses?

Q2: Given that SYMSEM is a modular framework and your VAR-LLM underperforms SOTA approaches like RESYM, why did you not integrate a SOTA model to maximize the framework's potential?

Q3: As noted in weakness 4, the SYMGEN paper already demonstrated and proposed the core idea of your "semantic-aware" approach. Can you please clarify the specific novelty of your method beyond what was already established by this prior work?

Q4: Can you provide SYMSEM's performance on more challenging datasets, such as Obfuscated Binaries, Malware, and IoT Firmware, to demonstrate its practical robustness?

---

### Official Review · Reviewer_WP3G · 2025-10-29

**Soundness:** 3
**Presentation:** 2
**Contribution:** 2
**Rating:** 4
**Confidence:** 4

**Summary:**

This paper presents SymSem, a self-transformative, semantic-aware fine-tuning framework for recovering function names from stripped binaries using LLMs. The framework employs a two-stage process. First, an LLM predicts variable names and types to enrich the decompiled code semantically. Second, an LLM is fine-tuned on this enriched code for name recovery. To address limited training data, the framework introduces a self-transformative data augmentation strategy that generates multiple semantic variants of each function by gradually replacing variable placeholders with ground-truth names during training. Evaluation across the x86-64, ARM, and MIPS architectures demonstrates that this framework yields better results than the state-of-the-art method SymGen, specifically achieving an F_1 score increase of up to 68% on MIPS.

**Strengths:**

+ The central two-stage concept of using an auxiliary model (VAR-LLM) to recover variable semantics addresses the semantic difference between symbolic training data and stripped testing data.

+ The self-transformative approach, which produces multiple semantically distinct function variants through controlled variable replacement, is sound for data expansion. This method avoids the risk of hallucination and the computational load associated with the summary creation employed in prior work.

+ The framework shows performance gains over baseline methods across diverse and underrepresented architectures.

**Weaknesses:**

- The paper reports a large performance difference for the baseline under its experimental setup without a thorough investigation or ablation study to account for this decline.

- The primary novel contribution relies on VAR-LLM, yet its performance is relatively low, as shown in Table 1: F1 scores reach a maximum of 0.263. In addition, the results compare only to a single external, non-contemporaneous approach (RESYM) that used a different dataset. This makes judging the absolute effectiveness difficult.

- Unclear limitations of the prior work on which the framework is built.

**Questions:**

The paper is promising in improving the state of the art, but the lack of clarity in the methodology and evaluations prevents full endorsement of whether the results will fully hold in practice.

(1) I observe that the VAR-LLM's low F_1 scores (maximum 0.263) are only weakly compared to external work RESYM on a different dataset. To position the VAR-LLM's performance rigorously, the paper would report variable recovery results for a dedicated variable recovery baseline (e.g., DIRTY or RESYM, if feasible) on the exact same dataset used.

In addition, given the VAR-LLM's low accuracy, the introduced noise might be substantial. The paper would include a control experiment where VAR-LLM's output is randomized (random name/type replacement) before FUNC-LLM input. If the actual VAR-LLM performance is not substantially better than this control, it would temper the claims about low-accuracy semantic recovery.

(2) To strengthen the motivation here, the specific drawbacks of SymGen (C1-C3) should be illustrated explicitly. For instance, to illustrate the general issue of function summaries being prone to hallucination, the authors can include a brief, fabricated example of a summary misinterpreting a function, e.g., describing a network function as a file I/O routine. As another example, to illustrate the semantic difference, the paper can provide a side-by-side example showing the input mismatch: meaningful names in training data (e.g., int read_config_file()) versus semantically void placeholders in testing data (e.g., int MASK(), using variables like v1).

(3) I observe a significant reduction in XFL's F_1 score from approximately 0.24 to 0.087 due to the change in data splitting, but this is insufficiently explained. The paper would provide a focused ablation study that shows how project-level versus function-level splitting specifically affects generalization for non-LLM baselines for fairness.

In addition, the optimal 25% replacement ratio is determined solely on the MIPS O3 setting. The paper would extend this ablation to at least one other architecture (e.g., x86-64) or optimization level (e.g., O0) to confirm the robustness of the ratio across diverse conditions.

(4) The paper would also clarify key technical concepts and the scope of the data scarcity challenge. First, the threshold and scoring mechanism used by CodeWordNet would be defined to measure partial correctness when matching generated subwords against ground truth. Second, the paper can explicitly list the size of the final, deduplicated, function-level training dataset (i.e., total number of unique functions) for each architecture to quantify the limited data challenge.

---

### Official Review · Reviewer_Zm4t · 2025-11-01

**Soundness:** 3
**Presentation:** 3
**Contribution:** 2
**Rating:** 4
**Confidence:** 3

**Summary:**

The authors propose SYMSEM, a semantic-aware, self-transformative fine-tuning framework for recovering function names from stripped binaries using LLMs. The approach aims to address limitations of current state-of-the-art methods like SYMGEN, which rely on LLM-generated function summaries that are prone to hallucination and introduce computational overhead.

SYMSEM operates through a two-stage pipeline: (1) fine-tuning a model (VAR-LLM) to recover variable names and types from decompiled code, and (2) fine-tuning a separate model (FUNC-LLM) to predict function names. The main contribution of the paper is a data augmentation strategy that generates multiple training variants of each function by progressively replacing stripped variable placeholders with ground-truth semantic information at different ratios.

**Strengths:**

1) The self-transformative data augmentation approach is innovative, generating multiple semantically meaningful training samples from single functions without requiring expensive LLM-generated summaries
2) The paper is well organized and well-presented
3) The approach achieves large improvement over some of the datasets.

**Weaknesses:**

My main concerns are evaluation-related:
1) The authors use a deterministic partitioning of the data (as they clearly point out). This prevents multiple runs and the use of statistical significance tests. As a result, it impossible to know whether the results are truly meaningful. Moreover, many of the differences reported in the results are less than 1%, which could easily be noise.

2) The paper fails to explain the dramatic performance differences across architectures (over 70% improvement for MIPS, much smaller for the others). Moreover, the authors' explanation is a bit strange: "This suggests that the base model is less familiar with underrepresented architectures such as MIPS, whose popularity has declined in recent years". If that is the case, and this dataset is underrepresented, how come your appraoch does better on it?

3) The paper provides no analysis of dataset characteristics (size, complexity, function name distributions) that differ across architectures, or anything else that can explain the changes in performance.

4) The analysis in the paper is rather limited, and does not address the points above. In the least, I would expect to see an analysis of the differences in function naming conventions across the dataset, or some other analysis that can explain the large differences in performance.

**Questions:**

Please refer to the weaknesses mentioned above.

---

### Note · Authors · 2025-11-12

I have read and agree with the venue's withdrawal policy on behalf of myself and my co-authors.